# Recognition Memory in Noonan Syndrome

**DOI:** 10.3390/brainsci11020169

**Published:** 2021-01-29

**Authors:** Floriana Costanzo, Paolo Alfieri, Cristina Caciolo, Paola Bergonzini, Francesca Perrino, Giuseppe Zampino, Chiara Leoni, Deny Menghini, Maria Cristina Digilio, Marco Tartaglia, Stefano Vicari, Giovanni Augusto Carlesimo

**Affiliations:** 1Child and Adolescent Psychiatric Unit, Department of Neuroscience, Bambino Gesù Children’s Hospital, IRCCS, 00165 Rome, Italy; paolo.alfieri@opbg.net (P.A.); cristina.caciolo@opbg.net (C.C.); paola.bergonzini@opbg.net (P.B.); deny.menghini@opbg.net (D.M.); stefano.vicari@opbg.net (S.V.); 2Center for Rare Diseases and Birth Defects, Department of Woman and Child Health, Institute of Pediatrics, Fondazione Policlinico Universitario Agostino Gemelli, IRCCS, Catholic University of the Sacred Heart, 00168 Rome, Italy; fraperrino1976@libero.it (F.P.); giuseppe.zampino@unicatt.it (G.Z.); leonichia2@gmail.com (C.L.); 3Rehabilitation Center UILMD Lazio Onlus, 00167 Rome, Italy; 4Genetics and Rare Diseases Research Division, Bambino Gesù Children’s Hospital, IRCCS, 00165 Rome, Italy; mcristina.digilio@opbg.net (M.C.D.); marco.tartaglia@opbg.net (M.T.); 5Medical Genetics, Academic Department of Pediatrics, Bambino Gesù Children’s Hospital, IRCCS, 00165 Rome, Italy; 6Department of Life Science and Public Health, Catholic University of the Sacred Heart, 00168 Rome, Italy; 7Laboratory of Clinical and Behavioral Neurology, Santa Lucia Foundation, 00179 Rome, Italy; memolab@hsantalucia.it; 8Department of Systems Medicine, Tor Vergata University, 00133 Rome, Italy

**Keywords:** RAS–MAPK, PTPN11, episodic memory, hippocampal memory processes, developmental disorders

## Abstract

Noonan syndrome (NS) and the clinically related NS with multiple lentiginous (NMLS) are genetic conditions characterized by upregulated RAS mitogen activated protein kinase (RAS–MAPK) signaling, which is known to impact hippocampus-dependent memory formation and consolidation. The aim of the present study was to provide a detailed characterization of the recognition memory of children and adolescents with NS/NMLS. We compared 18 children and adolescents affected by NS and NMLS with 22 typically developing (TD) children, matched for chronological age and non-verbal Intelligence Quotient (IQ), in two different experimental paradigms, to assess familiarity and recollection: a Process Dissociation Procedure (PDP) and a Task Dissociation Procedure (TDP). Differences in verbal skills between groups, as well as chronological age, were considered in the analysis. Participants with NS and NSML showed reduced recollection in the PDP and impaired associative recognition in the TDP, compared to controls. These results indicate poor recollection in the recognition memory of participants with NS and NSML, which cannot be explained by intellectual disability or language deficits. These results provide evidence of the role of mutations impacting RAS–MAPK signaling in the disruption of hippocampal memory formation and consolidation.

## 1. Introduction

Noonan syndrome (NS, OMIM PS163950) is a developmental disorder with an estimated incidence of from 1/1000 to 1/2500 live births [1], characterized by distinctive facies, postnatal growth retardation, cardiac defects, skeletal anomalies, and bleeding disorders, as well as mild developmental delay and/or learning difficulties [2,3,4].

NS is caused by germline mutations in genes coding for transducers and modulatory proteins with a role in the RAS–MAPK signaling cascade (i.e., PTPN11, SOS1, SOS2, KRAS, NRAS, RIT1, LZTR1, RAF1, and MEK1) [5,6], and, together with the clinically related NS with multiple lentiginous (NMLS, formerly called LEOPARD syndrome, OMIM PS151100), cardiofaciocutaneous syndrome, Costello syndrome and Noonan syndrome-like disorder with loose anagen hair, constitutes a group of known as RASopathies. The signaling pathway affected in RASopathies regulates the proliferation, survival and differentiation of cells [7,8], and is involved in synaptic function and the formation/consolidation of memory processes, particularly those which are hippocampus-dependent [9]. Several studies have frequently reported deficits in specific neuropsychological domains, such as learning and memory [10,11] in patients with NS and other RASopathies, with the heterogeneity in cognitive abilities being ascribed, at least in part, to the individual affected genes and type of mutation [12,13]. 

In animal models with dysregulated MAPK signaling, the impairment of mechanisms relative to long-term potentiation (LTP) and spatial learning has been shown [14,15,16,17]. Specifically, enhanced RAS-ERK activation was found to disrupt the balance between excitation and inhibition and to impair hippocampal LTP [18,19]. 

Notwithstanding increasing evidence that indicates that mutations impacting RAS–MAPK signaling can disrupt memory formation and consolidation, only a few studies have investigated declarative memory in individuals with NS and related disorders. In a previous study [20], we reported that children with NS associated with mutations in PTPN11, SHOC2, RAF1 or SOS1 display impairment more frequently in verbal free-recall tasks than in visual and spatial recognition memory tasks. Other authors found preserved verbal recognition memory but impaired recall performance in individuals with Costello Syndrome [21], a distinct RASopathy caused by mutation in the HRAS gene. Again, Pierpont, Tworog-Dube and Roberts [22] confirmed better performance in verbal recognition than in strategic free-recall tasks in NS. On the whole, available evidence indicates that in NS and related disorders the memory profile is characterized by impaired performance in recall tasks with an at least partially preserved performance in recognition memory tasks. 

One possible interpretation of these findings rests on the assumption that different memory experiences underlie performance in recall and recognition memory tasks. In particular, one hypothesis put forward is that recollection and familiarity, distinctively involved in recall or recognition memory tasks, may be affected differently in individuals with RASophaties [20,22]. 

In the last two decades, the dichotomy between recollection and familiarity experiences has been one of the most investigated issues in human declarative episodic memory. In recollection, exposure to a previously encountered item cues the retrieval of a number of contextual, temporal and other associative information that makes the experience of retrieval vivid, such as reliving the event in the mind’s eye. In familiarity, however, the item is recognized as having previously been encountered but does not cue any associative information, so the act of remembering is just a feeling of a previous encounter. The different characteristics of the two memory experiences make performance on recall tasks entirely dependent on recollection; conversely, recognition memory would be subsumed by either recollection or familiarity, the relative contribution of the two memory components depending at least partially on the associative vs. the single-item nature of the memorandum [23,24,25].

At a theoretical level, a critical question is what process or processes underlie these two experiences. According to dual-process theories [26,27], episodic retrieval depends on two independent processes: familiarity and recollection. 

In contrast, according to strength theories, episodic retrieval reflects a single memory strength dimension. A recent strength theory, the aggregate-strength model, incorporates the familiarity–recollection distinction by assuming that both processes are graded and are aggregated into a single strength signal before the retrieval decision [28,29,30,31,32,33]. More recently, The Complementary Learning Systems model proposes that recognition based on pattern completion is experienced as a generalized match with some prior experience, whereas recognition accompanied by awareness of relevant contextual details is based on pattern separation [34,35]. 

From a developmental perspective, there is documented evidence of age-related increases in tasks of memory recognition that require few contextual details and protracted development (i.e., gains continuing after middle childhood) than on tasks that require retention of detailed information [28,31,32,33,34,35,36,37,38,39,40,41,42,43]. Researchers have highlighted the importance of related cognitive functions, as working memory, metacognition and mnemonic strategies [44], in explaining individual differences in memory performance. Neuroimaging studies have shown a neural basis for age-related gains in memory performance. Differential developmental trajectories in the prefrontal cortex (PFC) and medial temporal lobe (MTL), including the hippocampus [41,42], have been linked to memory outcomes in associative memory test performance [45,46] and source memory performance [47,48]. Moreover, age-related differences in functional connectivity between PFC and MTL seem to account for age-related increases in memory performance [49].

On the other hand, impaired recollection has been documented in several developmental disorders, such as Autism spectrum disorder [50] and Williams syndrome [51], characterized by abnormal hippocampal and frontal functioning. Despite recollection and familiarity being amongst the most investigated issues in episodic memory, few studies have been conducted in young children or individuals with developmental disorders and genetic conditions and the issue is far from being completely understood. 

One reason for this may be related to the methods used to investigate such memory processes. Most of the evidence in this regard comes from studies that provided indirect estimates of recollection and familiarity by comparing performances in different tasks (for example, recognition tasks and free recall tasks). As concerns direct methods, some of them rely on an introspective assessment of the quality of remembering by the experimental subject (such as the Remember/Know paradigm and the Receiver operating characteristic), which are not easy to apply in young people or individuals with developmental disorders. Other methods, such as the Process Dissociation Procedure (PDP), provide more objective data that do not depend on introspection [25,27], but there have been very few studies using these kinds of methods in young populations with developmental disorders [50,51].

The aim of the current study was to provide a detailed characterization of recognition memory of children and adolescents with NS and clinically related NSML. To this end, we used two different experimental paradigms which more directly assess the contribution of familiarity and recollection to recognition memory tasks. The first was a modified version of the Process Dissociation Procedure (PDP) devised by Jacoby [52] and adjusted for a developmental population by Costanzo et al. [51]. In this paradigm, the recollection and familiarity components of recognition are estimated by contrasting different recognition conditions for the studied material [40,52,53] under different source memory conditions. The second was a Task Dissociation Procedure (TDP), where performance on tests of single-item recognition, which reflect familiarity strengths, is compared to performance on tests of associative item recognition, which requires retrieval of contextual information involved in recollection. 

Consistent with previous evidence of lower performance on recall than recognition tasks, we expected that individuals with NS recollection would be more affected on recollection than familiarity components of recognition memory. Moreover, the two experimental paradigms were also adjusted to test the hypothesis of a possible effect of the modality of memorandum, since previous findings in NS and other RASopathies displayed better performance in verbal than in visual memory tasks [20,22]. Indeed, the material used in the PDP included words or easily verbalizable pictures. Conversely, the memorandum in the TDP was only pictorial and hardly verbalizable.

## 2. Materials and Methods

### 2.1. Participants

Eighteen children with molecularly confirmed NS (*N* = 15; F/M = 8/7) or clinically related NSML (*N* = 3; F/M = 2/1) participated in the study. Participants (mean chronological age—CA = 8.74 years, SD = 2.95, range 5.42–13.75 years) were compared with 22 typically developing (TD) children (F/M = 14/8; mean CA = 8.22 years, SD = 1.57, range 6.08–11.25 years) matched for chronological age and non-verbal IQ. Children with NS/NSML were part of a larger pool of individuals attending the Bambino Gesù Children’s Hospital IRCCS and Agostino Gemelli Hospital IRCCS, both in Rome, for a clinical and rehabilitative follow-up. Diagnosis was made by experienced medical geneticists and pediatricians based on clinical evaluation. In all individuals, diagnosis was confirmed by molecular analysis (PTPN11, *N* = 11; SOS1, *N* = 5; RAF1, *N* = 1; RIT1, *N* = 1) (see Table 1 for details).

Selection criteria for recruitment into the study included IQ in the average range and normal or corrected-to-normal vision and hearing as reported in chart notes. Exclusion criteria were reports of neurological signs and the presence of ADHD or other psychopathological disorders. The children in the control group were recruited from local schools. Their parents reported them to be in good health. Exclusion criteria were reports of neurological signs, history of language delay or learning disability and presence of ADHD or other psychopathological disorders. Observations were carried out after informed consent had been obtained from all participants and their families and the study had been approved by the Bambino Gesù Children’s Hospital IRCCS ethical committee (Code: 1412_OPBG_2017). The authors assert that all procedures contributing to this work comply with the ethical standards of the relevant national and institutional committees on human experimentation and with the Helsinki Declaration of 1975, as revised in 2008.

Participants’ cognitive level was assessed using Raven’s Colored Progressive Matrices [54]. Mean IQ was 105.44 (SD = 14.55, range 80–130) in the NS and NSML group and 108.14 (SD = 11.23, range 87–130) in the TD group. The lexical quotient (LQ) was assessed using the Peabody Picture Vocabulary Test (PPVT) [55] which measures lexical comprehension. The two groups did not differ for both CA (Welch’s F(1,25) = 0.44, *p* = 0.51, pη^2^ = 0.01) and non-verbal IQ (F(1,38) = 0.44, *p* = 0.51, pη^2^ = 0.01), but they differed for LQ (F(1,38) = 7.09, *p* = 0.01, pη^2^ = 0.16). Mean LQ in the NS/NSML group (93.61; SD = 15.48; range 74–122) was lower than in the TD group (104.45; SD = 10.15; range 87–122). Since participants differed for LQ, we entered this variable as a covariate in the analysis. 

### 2.2. Experimental Tasks

The memory tasks based on the two memory paradigms, namely the PDP and TDP, were given the same day with an interval of about ten minutes between them in a counterbalanced order between participants. The entire memory evaluation session lasted approximately 40 min. 

#### 2.2.1. Process Dissociation Procedure

The PDP used here was adapted for a developmental population in our previous study [30]. Participants have to encode and successively recognize two blocks of items, one presented visually and one in an auditory format. When tested later, they were asked to indicate whether or not they saw the item earlier, after one of the two recognition conditions. In the inclusion condition, participants were instructed to recognize as “old” all previously presented items, regardless of the list on which they had seen them, and to reject the novel items. In the exclusion condition, participants were instructed to recognize only the items presented visually as “old”, and to reject those presented in the other list as well as the novel items. Due to the differential request of remembering the source list to perform the recognition task in the two conditions, it is acknowledged that recollection plays a more crucial role in the exclusion than the inclusion condition of the PDP. Therefore, the contrast of hits and false-alarm responses in the inclusion and exclusion conditions allows the direct estimation of the contribution of recollection and familiarity to recognition performance.

##### Stimuli

The experimental material in the test phase consisted of 90 items selected from the Snodgrass and Vanderwart [56] set of black-and-white line drawings. Frequency of occurrence of the items ranged from 40% to 60% in the Italian children’s first words vocabulary [57]. There was a colored drawing for each of the 90 items (visually different from Snodgrass and Vanderwart‘s drawings), representing a typical example of the item and the word name (spoken aloud by the examiner). The 90 items were split into two lists, which were comparable in terms of mean frequency of occurrence in the Italian children’s first words vocabulary. One of the lists was assigned for the inclusion condition of PDP and the other for the exclusion condition. The list assignment to each experimental condition was counterbalanced across participants. Example of items of the PDP are shown in Figure 1.

##### Procedure

Throughout the experiment, the participants were seated in a comfortable chair facing a 15” portable PC screen located on a table about 40 cm away from them. Two recognition memory tests were given, one with inclusion instructions, the other with exclusion instructions. Although instructions for the study phase were identical for the inclusion and exclusion condition, instructions differed for the test phases of the two conditions [37]. Each study phase consisted of two blocks of stimuli: participants were shown one set of 15 colored drawings and they then listened to a set of 15 words spoken aloud by the examiner. In both cases, they were instructed to try to remember the presented stimuli for a subsequent memory test. The participants’ task was to name each item and then to make a yes/no judgment about pleasantness. In the auditory block, the words were spoken aloud by the examiner at a rate of one word every 2 s. The participants were asked to repeat each word aloud and again to make a yes/no judgment about pleasantness. In order to increase the likelihood of the children being able to discriminate between the two stimulus sources, the screen had a white background during presentation of the colored drawing and a blue background during auditory presentation of the words. Thus, the participants could discriminate between the two blocks based on stimulus format (colored drawing vs. spoken word), background color (white vs. blue), and order of presentation (first vs. second study block).

During the test phase (immediately after the study phase), the participants were shown the overall set of 45 black-and-white line drawings (corresponding to 30 studied and 15 unstudied items) in random order. Each drawing was presented at the center of the PC screen (covering about 20° of visual field) for 2 s, with an inter-trial interval of approximately 2 s. 

In the inclusion condition, which was given after the first study phase, participants were requested to respond “old” to all items presented in the study phase, irrespective of the modality (visual or auditory) of presentation, and to respond “new” to unstudied items. Conversely, in the exclusion condition, which was given after the second study phase, participants were asked to respond “old” only to items that had been studied as colored drawings and to respond “new” to items studied in the auditory modality and to unstudied items. 

A 5-min pause separated the recognition memory task with inclusion instructions from the study phase of the exclusion task. No feedback was given after the participants responded. To ensure that all participants fully understood what they were required to do, a practice phase, including both study and test blocks, was run before the start of the study phase of both the inclusion and exclusion tasks. Six items (different from those in the experiment) were used. If a subject failed on more than two items in one of the two practice runs, he was excluded from the study. See Costanzo et al. [30] for further details.

#### 2.2.2. Task Dissociation Procedure

The experimental paradigm contrasted participants’ performances on tests of single-item and associative recognition of visual stimuli. Indeed, if familiarity reflects the memory strength of single items, then it should be effective at discriminating between studied and not-studied items, as in a test of item recognition, but it should be less helpful in the retrieval of contextual and other associative information, as is required in tests of associative recognition.

##### Stimuli

The experimental material consisted of 63 close-up color photos of flowers and nine color photos of butterflies with spread wings. Only photos of flowers and butterflies that could not be named in a pilot study were included in the experimental material. Moreover, many items were of the same or very similar colors in order to make encoding of items based on color verbalization impossible. The 63 photos of flowers served as stimulus material for the single-item recognition task. Nine photos were used as study items and the remaining 54 were used as distracters in the recognition test. The same nine photos of flowers studied in the single-item recognition task and the nine colored photos of butterflies were used in the associative recognition task. Example of items of the TDP are shown in Figure 2.

##### Procedure

The experimental setting included two memory tasks: first, the single-item recognition task, then the associative recognition task. 

Each task consisted of three learning trials. In the study phase of the single-item recognition task, participants were presented with nine flowers. Items appeared singly at the center of the PC screen (covering about 15° of the visual field) on a white background for 3 s. For each item, participants had to make a yes/no pleasantness judgment and were asked to remember the items for a successive memory test. Immediately after the study phase, nine triplets of flowers were presented; each triplet consisted of a previously studied flower (target item) and two new flowers presented vertically. The associative recognition task was administered immediately after the single-item recognition task. The study phase consisted of the presentation of nine pairs, each including a butterfly and a flower photo. Each pair was presented individually for 3 s; the butterfly appeared in the upper half of the screen and the flower in the lower half, each photo covering about 15° of the visual field. Participants were requested to make a yes/no pleasantness judgment about each pair and were encouraged to associate the two items in each pair and try to remember them for a subsequent memory test. Each recognition item included a butterfly and a flower triplet composed of the flower that had been presented together with the probe butterfly during the study phase and two others that had been presented together with two different butterflies during the study phase. The participants were requested to indicate the flower that had been presented with the butterfly during the study phase. By requiring the participants to select the associated item from among other previously studied items, we were confident that their decision could not be based on a single-item familiarity judgment but would necessarily involve remembering the between-item association. As in the single-item recognition task, two additional trials, each of which had a study and a test phase, followed. For both the single-item and the associative recognition task, the order of the three sets of test items was counterbalanced across participants. See Costanzo et al. [30] for further details.

### 2.3. Statistical Analysis

Comparison on demographic characteristics between groups were performed by one-way ANOVAs, while the Chi-squared test was used to value the non-parametric variables. Raw data from the PDP, specifically the proportion of “old responses”, were analyzed by means of repeated measures ANCOVA with Group (NS vs. TD) as a between factor, Condition (inclusion vs. exclusion) and Stimuli (colored drawings vs. spoken words) as within factors, and LQ and CA as covariates as covariates. By applying the dual-process signal detection model to the results of the PDP, one-way ANCOVAs were conducted to compare the parameter estimates of recollection and familiarity between groups, with LQ and CA as covariates. Data from the single-item and associative recognition task were analyzed by a repeated measures ANCOVA, with Group (NS vs. TD) as between factor, Task (single items vs. associations) and Trial (from I to III) as within factors and LQ and CA as covariates. 

Levene’s test was used to check for homogeneity and Mauchly’s test was used to check for a violation of sphericity. The overall level of significance was set at *p* < 0.05. The Welch test was performed on demographic characteristic comparisons when the homogeneity of variance of the data was invalid. If a violation of sphericity was detected, a Greenhouse–Geisser correction was applied. In the ANCOVAs analyses, since CA had a skewed distribution, data were normalized by by Box–Cox transformation. Post hoc analyses were performed using the Tukey‘s test. Effect sizes of ANOVAs were given in partial eta-squared (pη^2^).

## 3. Results

### 3.1. Process Dissociation Procedure 

The proportion of “old” responses to studied and new items made by participants with NS and TD in the inclusion and exclusion conditions of the PDP task are summarized in Table 2. 

In the inclusion condition, the probability of recognizing an item as “old” (target) reflects hit rates for the items presented at study as spoken words or colored drawings, whereas the false alarm rate reflects “old” responses to unstudied items. In the exclusion condition, the probability of recognizing an item as “old” only reflects hit rates for items presented at study as colored drawings, whereas the false alarm rate reflects “old” responses for both items studied as spoken words and as new items. 

To establish whether assumptions underlying the PDP were met, it was first examined whether acceptance rates of new items differed between inclusion and exclusion across groups. The proportion of “new” responses for the colored drawings and spoken words were analyzed by means of repeated measures ANCOVA with Group (NS vs. TD) as a between-factor and Condition (inclusion vs. exclusion) as within factors, and LQ and CA as covariates. The Group (F(1,36) < 0.001; *p* = 0.99, pη^2^ < 0.01), the Condition (F(1,36) = 1.62; *p* = 0.21, pη^2^ = 0.04) and the interaction between Group and Condition (F(1,30) = 0.76; *p* = 0.39, pη^2^ = 0.02) all did not produce significant results. The proportion of “old” responses for the colored drawings and spoken words were analyzed by means of repeated-measures ANCOVA with Group (NS vs. TD) as a between-factor, Condition (inclusion vs, exclusion) and Stimuli (colored drawings vs. spoken words) as within factors, and LQ and CA as covariates. The Group (F(1,36) = 1.45; *p* = 0.24, pη^2^ = 0.03), the Condition (F(1,36) = 0.91; *p* = 0.35, pη^2^ = 0.02) and the Stimuli (F(1,36) = 2.26; *p* = 0.14, pη^2^ = 0.06) effects, nor the interaction between Group and Stimuli (F(1,36) = 2.20; *p* = 0.15, pη^2^ = 0.08) and the triple interaction between Group, Condition and Stimuli (F(1,36) = 1.22; *p* = 0.28, pη^2^ = 0.06) all did not produce significant results. Conversely, the interaction between Group and Condition was significant (F(1,36) = 4.71; *p* = 0.04, pη^2^ = 0.12), due to the higher proportion of “old” responses in the TD compared to the NS (*p* < 0.001) group in the inclusion condition, but not in the exclusion condition (*p* = 0.99).

Recollection was calculated by subtracting “old” responses to non-targets (spoken words) in the exclusion condition from “old” responses to non-targets in the inclusion condition, R = “old” responses inclusion_nontarget_—“old” responses exclusion_nontarget_. Familiarity was calculated from estimates of recollection, F = “old” responses exclusion_non-target_/(1 − R) (see [40,52,53,54,55]).

The one-way ANCOVAs revealed no significant difference among groups for the familiarity estimate F (participants with NS/NMLS = 0.24 ± 0.30; TD = 0.26 ± 0.36; F(1,36) = 0.52, *p* = 0.48, pη^2^ = 0.01), but TD displayed a significantly higher recollection estimate R (0.88 ± 0.13) than participants with NS/NMLS (0.72 ± 0.22; F(1,36) = 5.24, *p* = 0.03, pη^2^ = 0.13).

### 3.2. Task Dissociation Procedure

Table 3 reports the average proportion of accuracy of participants with NS/NMLS and TD on the single-item and associative recognition task per each trial. A repeated-measures ANCOVA, with Group as between-factor, Task (single items vs. associations) and Trial (from I to III) as within factors, and LQ and CA as covariates, revealed a significant main effect of Group (F(1,36) = 6.04, *p* = 0.02, pη^2^ = 0.14). In fact, across the two tasks TD participants were, on average, more accurate than participants with NS/NMLS. The Task effect was also significant (F(1,36) = 4.15, *p* = 0.05 after Greenhouse–Geisser correction, pη^2^ = 0.10) because of the higher average accuracy obtained by all participants on the single-item compared to the associative recognition test. A general Trial effect did not emerge (F(2,72) = 1.69, *p* = 0.19 after Greenhouse–Geisser correction, pη^2^ = 0.05). 

Interestingly, the Group × Task interaction was significant (F(1,36) = 3.30, *p* = 0.03 after Greenhouse–Geisser correction, pη^2^ = 0.13). Post hoc analyses showed that participants with NS/NMLS (mean percentage of accuracy 0.95 ± 0.09) and controls (mean percentage of accuracy 0.98 ± 0.09) performed at the same level on the single-item recognition test (Tukey test, *p* = 0.89), whereas the participants with NS/NMLS scored significantly lower (mean percentage of accuracy 0.48 ± 0.17) than TD (mean percentage of accuracy 0.63 ± 0.14) on the associative recognition task (Tukey test, *p* < 0.001). This result indicates that participants with NS had particularly low performance in remembering associated items but not single items. The Group × Task × Trial interaction (F(2,72) = 0.02, *p* = 0.97 after Greenhouse–Geisser correction, pη^2^ < 0.01) as well as the Task × Trial interaction (F(2,72) = 0.91, *p* = 0.39 after Greenhouse–Geisser correction, pη^2^ = 0.02) were not significant.

## 4. Discussion

The aim of the present study was to evaluate the contribution of familiarity and recollection to the recognition memory of children and adolescents with NS and clinically related NSML. For this purpose, a cohort of 18 children with molecularly confirmed diagnosis was compared with a group of TD children matched for chronological age and non-verbal IQ, on two different experimental paradigms. The first paradigm allowed for direct estimation of the recollection and familiarity components of recognition by setting the two processes in opposition in the inclusion and exclusion conditions of a PDP task [40,52,53], which is similar to a source memory discrimination. The second paradigm compared the performance of participants on a memory task mainly relying on familiarity (single-item recognition) with performance on a memory task requiring a larger contribution of recollection (between-item associative recognition) [25]. The two experimental paradigms differed as regards the nature of the memorandum, which was verbal (or easily verbalizable pictures) in the PDP paradigm, and pictorial and hardly verbalizable in the single-item and associative recognition task.

The results of the two experimental tasks consistently demonstrated a reduced contribution of detailed memory, or recollection, and a preserved contribution of familiarity, or memory which requires few contextual details, to the recognition performance of participants with NS/NSML. Indeed, in the PDP task, our participants were as accurate as TD in recognizing previously studied stimuli, but they failed to remember the modality in which the items had been studied. Consequently, the two groups displayed similar estimates of familiarity but the estimate of recollection in the NS/NSML group was significantly reduced compared with that of the TD group. 

Performance on the single-item recognition task of the TDP confirmed that individuals with NS/NSML were as accurate as controls in identifying recently seen stimuli. We acknowledge that the virtually perfect performance of the two groups in the single-item recognition task of the TDP (mean percentage of correct responses ≥0.95 for each of them) may represent a limitation in revealing a clear-cut dissociation between groups in this task. Specifically, the presence of only three retrieval trials may be prevent the possibility of detecting a possible difference between participants in the single-item task. However, on the associative recognition task, participants with NS/NSML were significantly poorer than controls in identifying the visual stimulus that had been associated with a stimulus target during the study phase. 

Consistent with the results on the PDP, we interpreted these results as evidence of the preserved contribution of familiarity and the reduced contribution of recollection to the memory performance of individuals with NS/NSML. Indeed, because the participants had to discriminate between previously seen and unseen flowers in the single-item recognition test, familiarity processes probably played a prevalent role in the performance of this task. Furthermore, the use of novel flowers enhances the salience of familiarity because previously studied items are more familiar than new items. On the other hand, since in the associative recognition task the alternative choices (flower photos) were all from the studied material, the participants’ decision must have involved recollection of the previously studied associations and details. 

The fact that the two groups differed as regards to verbal comprehension abilities might partially explain these results, given that at least one procedure (PDP) required verbal processing of memorandum. However, also considering differences in verbal ability in the analysis, the results substantially hold, showing that the impaired recollection of these individuals is unrelated to the presence of poor language abilities. 

Moreover, given the well-recognized age-related differences in multiple hippocampal subfield volumes in the age range of our patients [19,43,45,46,58], we considered our results taking into account possible change related to age.

To our knowledge, this is the first study that reports evidence from direct recognition memory procedures of a relatively preserved generalized recognition memory and impaired detailed memory of a group of individuals with NS/NSML. Previous evidence from the literature has only been indirect [20,22], based upon the contrast between performances on tests of free recall (believed to recruit only recollective processes) and on recognition (which is assumed to involve both recollection and familiarity). 

From a developmental perspective, there is evidence that recollection emerges later and shows more developmental changes than familiarity [36,37,38,39,40]. Moreover, impaired recollection and preserved familiarity has been documented in other developmental disorders such as Autism spectrum disorder [50] and Williams syndrome [51], characterized by abnormal hippocampal functioning.

From a neurobiological perspective, the experimental investigation into the neural underpinnings of recollection and familiarity has mainly been concentrated on archi- and neo-cortical structures in the mesial temporal lobe (hippocampus and parahippocampal gyrus, respectively). According to dual-process theories [26,27], at the level of the mesial temporal lobes, the hippocampus proper supports the cognitive operations necessary for recollection memory, whereas various cortical areas in the parahippocampal gyrus (especially the perirhinal cortex) would be implicated in familiarity detection [26,59]. Following the assumptions of the dual-process theories, our finding of reduced recollection and spared familiarity in individuals with NS would suggest a differential involvement of cortical regions in the mesiotemporal lobe. In particular, while the hippocampus, crucially involved in recollection, should be the site of neurodevelopmental delay, the associative cortical regions in the parahippocampal gyrus, which support familiarity memory judgments, should be substantially spared. 

Neurobiological data in support of a particularly impaired hippocampal functioning in NS individuals would be related to the evidence of aberrant hippocampal neuronal activity-induced signalling in RASopathies. Indeed, the activation of the RAS-MAPK pathway is involved in triggering long-term synaptic changes in the CA1 subfield [14,16,17] which, in turn, has been implicated in the regulation of AMPARs delivery to the postsynaptic plasma membrane [60].

Many investigators have proposed that the lack of detailed memory for early childhood memories is due to the protracted development of the hippocampus [61,62]. Moreover, recent studies have shown age-related differences in multiple hippocampal subfield volumes from 4 to 13 years of age [45,46,58]. A possible related hypothesis is that recollection deficits in children with NS may be related to a delayed developmental time course of hippocampal formation and connectivity.

A recent neuroimaging study reported data in NS that are at least partially inconsistent with this view. Indeed, Johnson et al. [63] reported no differences in hippocampal volumes between participants with NS and healthy controls; conversely, they found cortical abnormalities in the NS group at the level of bilateral entorhinal regions and right parahippocampal gyrus. Regarding the lack of morphological alterations in the hippocampus, recent evidence in the NS-related Ptpn11D61Y mouse model suggests that aberrant neuronal-activity-induced signaling is not necessarily reflected in changes in neuronal morphology [64]. Indeed, in these animals, no macroscopic volume abnormality of the hippocampus, but possible synaptic changes in the afferent/efferent fibers, was observed.

The alternative theoretical position relative to the nature of memory processes involved in familiarity and recollection assumes, instead, that episodic retrieval reflects a single memory strength dimension. As for the neural circuits implicated, this theory does not posit a differential involvement of mesial temporal lobe structures in recollection and familiarity, since both the hippocampus and perirhinal cortex would be implicated in the memory strength detection underlying the two components [31,32,33,65].

Following the memory strength hypotheses of recollection and familiarity, a widespread involvement of mesiotemporal structures, involving both the hippocampus and parahippocampal cortices, in individuals with NS, is expected. Johnson et al. [63] did not report the exact localization of atrophic changes in the parahippocampal gyrus of NS individuals. Fattah et al. [66] found that children with NS exhibited significant decreases in the volume and thickness of striatum and regions surrounding the hippocampus. Specifically, the bilateral entorhinal cortex exhibited significant decreases, as did the right temporal pole surface area and the right parahippocampal gyrus.

The hypothesis of greater involvement of the hippocampus in the recognition memory deficits of individuals with NS is, however, strengthened by the specific low performance shown by our participants with NS in the experimental conditions assessing associative recognition. Indeed, the PDP could be considered a test which, in the condition of exclusion, investigates source memory skills, a type of associative memory in which the correct recollection of the context is a prerequisite for adequate performance. The task dissociation procedure, in turn, explicitly contrasts a condition of recognition of single items and a condition of recognition of between-item associations. There is general consensus in the literature that associative memory recruits hippocampal resources, whereas single-item recognition can be accomplished by parahippocampal cortices [67,68]. In particular, compelling evidence has been provided of a relevant role of hippocampus in source memory as well as recall and associative memory; for a review, see [69].

In children with NS, alterations of the cortical regions adjacent to the hippocampus [63] specifically affected the connectivity of these regions; in particular, a reduction in surface area of the gray–white boundary was found in bilateral entorhinal regions and a reduction in cortical thickness was found in the right parahippocampal gyrus. Parahippocampal cortices, including parahippocampal place area and the fusiform gyrus, have also been involved in visual recognition processing, specifically in object–object or object–context relational processing [70,71]. Given that the material used in the TDP was only pictorial and required object–object relational recognition, it cannot be excluded that the associative recognition deficit, exhibited by children with NS, could partly be due to perceptual deficits per se, related to parahippocampal abnormalities. Previous findings in NS and other RASopathies have indeed displayed better performance in verbal than visual memory tasks [20,22]. The fact that children with NS have also shown source memory deficits in the PDP, which included words or easily verbalizable pictures, suggests a more generalized deficit in detailed episodic memory in NS population, beyond the modality of the memorandum. This is in line with fMRI evidence [72], who found a degree of specialization, with respect to stimulus type, within the MTL structures. Indeed, different parahippocampal cortices underlie familiarity for different kinds of visual stimuli, while the hippocampus mediated recollection, regardless of the kind of stimulus. 

A reduction in cortical thickness was also reported in the middle frontal areas in children with NS [63,66]. One could hypothesize that abnormalities in the connectivity of the cortical regions adjacent to the hippocampus (with frontal and hippocampal cortices), rather than their integrity, might explain the memory profile of impaired associative recognition and preserved general memory strengths in our group of children and adolescents with NS/NSML. 

A slower development of memory for context has been associated with the maturation of PFC functioning [42]. Postmortem and structural imaging evidence indicates that PFC maturation appears to continue into late adolescence [49,73]. For this reason, the role of reduced prefrontal white matter maturation in children with NS, as well as reduced hippocampal contributions, may account for the associative memory deficits found in our group of children with NS/NSML. Taken together, findings of age differences in PFC activation patterns suggest that the development of attentional and strategic control processes involved in the encoding and retrieval of detailed representations of experiences is key to predicting age-related differences in memory performance.

As an alternative framework for interpreting the impaired performance on recognition memory tasks in individuals with NS/NSML, the role of extra-memory cognitive abilities should be considered. Indeed, as widely demonstrated in the literature on typical individuals, recollection and familiarity processes are affected differently by a number of experimental variables For example, the consistency of the perceptual format of stimuli during study and testing is more critical for familiarity than recollection [72,74]. 

Conversely, the reduced availability of attentional resources affects estimates of recollection much more than estimates of familiarity [75,76]. Given that perceptual alterations [77,78] and attention problems [79,80] have been reported in individuals with NS/NSML, such variables are potentially relevant for interpreting the qualitative aspects of performance in the recognition memory task of our cohort of children and adolescents with NS and NSML, although the presence of ADHD was considered an exclusion criterion in our study. It could be hypothesized that in individuals with NS/NSML, the reduced efficiency of some control mechanisms (e.g., those related to attentional or executive abilities) might interfere with the deep encoding and/or strategic retrieval of target stimuli in a way that impairs recollection more than familiarity. Certainly, further brain-behavioral studies are needed to clarify the specific neurobiological mechanism underpinning the dissociation between general strength memory and associative recognition memory in individuals with NS and NSML, as documented in the present study and in previous reports [20,22].

The impairment of detailed and associative recognition memory can have important clinical implications. The awareness of recollection impairment could direct therapists toward a more efficient, process-oriented cognitive treatment of NS learning disabilities. Indeed, manipulating the material to be learned in order to increase the reliance on familiarity (spared) rather than associative memory (impaired) (e.g., by increasing consistency of the perceptual format of stimuli during study and testing) or by repeating or spacing learning sessions, as demonstrated in animal models [81], could improve learning therapy for individuals with NS.

## 5. Conclusions

The study aimed to provide a detailed characterization of the contribution of familiarity and recollection components to the recognition memory of children and adolescents with NS/NSML. By using two independent and direct experimental paradigms, our study shows a reduced contribution of recollection in the recognition memory performance of a group of children and adolescents with NS and the clinically related NSML. The impairment of detailed memory or recollection in children with RASophaties is direct evidence supporting the neurobiological hypothesis that RAS–MAPK mutations can disrupt hippocampal and temporal lobe memory formation and consolidation. The present findings provide insights into the declarative memory profile of these individuals, enabling the provision of reliable strategies and methods for an efficient rehabilitation of their learning and memory deficits. Moreover, this specific memory deficit in individuals with NS/NSML could be the target of potential new treatments, for example the use of inhibitors of the RAS-MAP-ERK cascade, as has already been attempted in animal models to rescue cognitive deficits.

## Figures and Tables

**Figure 1 brainsci-11-00169-f001:**
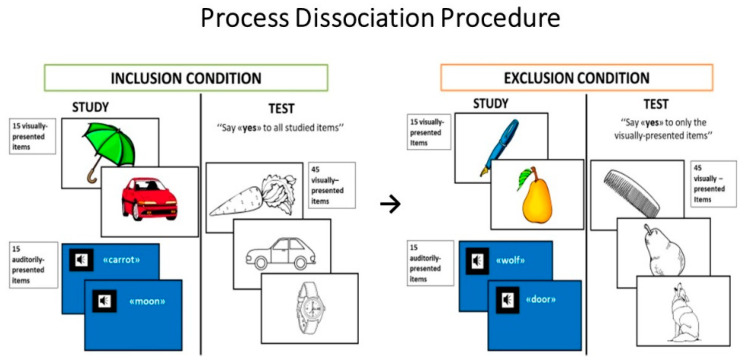
Examples of items in the Process Dissociation Procedure. Two recognition memory tests were given: one with inclusion, the other with exclusion instructions in the test phase. Although the study phase was identical, instructions differed for the test phases of the two conditions. Each study phase consisted of two blocks of stimuli: one set of 15 colored drawings and a set of 15 words spoken aloud by the examiner. During the test phase (immediately after the study phase), the participants were shown a set of 45 black-and-white line drawings (corresponding to 30 studied and 15 unstudied items) in random order. In the inclusion condition, participants are instructed to recognize as “old” all previously presented items, regardless of the list on which they had seen them, and to reject the novel items. In the exclusion condition, participants are instructed to recognize as “old” only the items presented visually, and to reject those presented in the other list as well as the novel items.

**Figure 2 brainsci-11-00169-f002:**
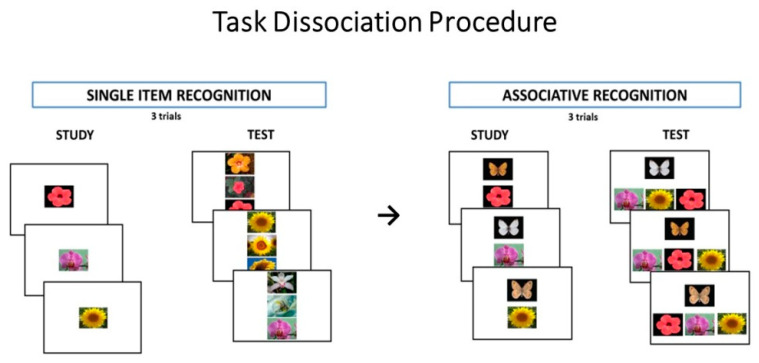
Examples of items in the Task Dissociation Procedure. The experimental setting included two memory tasks: the single-item recognition task and the associative recognition task. Each task consisted of three learning trials. In the study phase of the single item recognition task, participants were presented with 9 flowers. Immediately after the study phase, 9 triplets of flowers were presented; each triplet consisted of a previously studied flower (target item) and two new flowers presented vertically. The associative recognition task was administered immediately after the single item recognition task. The study phase consisted of the presentation of 9 pairs, each including a butterfly and a flower photo. Each recognition item included a butterfly and a flower triplet composed of the flower that had been presented together with the probe butterfly during the study phase, and two others that had been presented together with two different butterflies during the study phase. The participants were requested to indicate the flower that had been presented with the butterfly during the study phase.

**Table 1 brainsci-11-00169-t001:** Molecular characterization of the study cohort. Legend. F = female, M = male, IQ = Non-verbal intellectual quotient, LQ = Lexical quotient, NS = Noonan Syndrome, NSML = Noonan syndrome with multiple lentiginous.

Subject	Sex	Age	IQ	LQ	Disorder	Gene	Amino Acid Change
1	M	6.08	90	84	NS	*SOS1*	Gly434Arg
2	F	13	90	90	NS	*PTPN11*	Asn308Ser
3	F	6.75	120	98	NS	*PTPN11*	Thr42Ala
4	F	8.58	100	106	NS	*SOS1*	Ile733Phe
5	M	6.42	120	84	NS	*PTPN11*	Ala72Ser
6	M	10.08	102	94	NS	*RIT1*	Ser35Thr
7	F	12.5	80	95	NS	*SOS1*	Arg552Lys
8	M	13.75	100	111	NS	*PTPN11*	Met504Thr
9	F	5.75	90	84	NS	*SOS1*	Arg552Ser
10	M	6.25	110	89	NS	*PTPN11*	E139D
11	F	5.75	130	102	NS	*SOS1*	Glu433Lys
12	M	7.33	100	71	NS	*PTPN11*	Asn308Ser
13	F	5.41	110	74	NS	*RAF1*	S257L
14	F	7.66	90	73	NS	*PTPN11*	Phe285Ser
15	M	13.08	120	80	NS	*PTPN11*	Gly503Arg
16	F	7.42	101	120	NSML	*PTPN11*	Tyr279Ser
17	F	12.33	125	122	NSML	*PTPN11*	Thr468Met
18	M	9.08	120	108	NSML	*PTPN11*	Thr468Met

**Table 2 brainsci-11-00169-t002:** Average (SD) proportion of “old” response to the Processing Dissociation Procedure.

Type of Item	Inclusion Condition	Exclusion Condition
NS/NMLS		
Colored drawings	0.87 (0.19)	0.87 (0.19)
Spoken words	0.80 (0.17)	0.08 (0.08)
New	0.03 (0.05)	0.02 (0.04)
TD		
Colored drawings	0.92 (0.12)	0.89 (0.13)
Spoken words	0.93 (0.08)	0.05 (0.09)
New	0.04 (0.07)	0.03 (0.05)

NS/NMLS = children Noonan Syndrome/clinically related Noonan Syndrome with multiple lentiginous, TD = Typically developed children.

**Table 3 brainsci-11-00169-t003:** Average (SD) proportion of accuracy of participants with NS/NMLS and TD on the single-item and associative recognition task per each trial.

Task	NS/NMLS	TD
SINGLE ITEM		
Trial 1	0.93 (0.12)	0.95 (0.14)
Trial 2	0.96 (0.08)	0.98 (0.03)
Trial 3	0.96 (0.07)	1.00 (0.00)
ASSOCIATIVE RECOGNITION		
Trial 1	0.40 (0.15)	0.49 (0.22)
Trial 2	0.50 (0.22)	0.64 (0.20)
Trial 3	0.54 (0.25)	0.75 (0.14)

NS/NMLS = children Noonan Syndrome/clinically related Noonan Syndrome with multiple lentiginous, TD = Typically developed children.

## Data Availability

The data that support the findings of this study are available on request from the corresponding author. The data are not publicly available due to privacy or ethical restrictions.

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
