# Peer review of "Recognition Memory in Noonan Syndrome"

_brainsci, 2021, doi:10.3390/brainsci11020169_

Round 1
Reviewer 1 Report
The manuscript reports a study that examined long-term memory (LTM) deficits in Noonan Syndrome (NS) youth, relative to normally developing youth. The mean ages of the NS and healthy groups were approximately 8 to 9 years of age, meaning that the experiments assessed LTM capability of the participants well before maturation of their functional hippocampal-cortical brain networks, which are necessary for the full fidelity in LTM retrieval (see Ofen et al., 2007, Nature Neuroscience).
The experiment groups are appropriately characterized, the methods well explained and the results clear. The Introduction of the study and Discussion of the findings are useful, if quite lengthy. The English translation can improve in syntax and noun-verb agreement. The two LTM measures are appropriate for the authors’ motivation to assess the participants’ high-fidelity LTM capability, but the authors’ interpretations according to the Yonelinas high-threshold dual process (HTDP) model do not serve their findings adequately.
Rather than struggle with the known and demonstrated limitations of the HTDP model beheld in the text, the report could bring more valuable insight to the literature about NS LTM deficits if it were to draw a more balanced view of cognitive models in high-fidelity LTM. Three major and a few minor concerns are explained below.
- The manuscript refers to recollection and familiarity, from line 2 through line 546, as though the model for these catch-all memory process labels were well defined and accepted in the relevant literature. Quite to the contrary, a hot controversy about the substantial limitations of HTDP has roiled the field (Slotnick & Dodson, 2005, Memory & Cognition; Wixted, 2007, Psychological Review; Wais, 2008, Neuropsychologia). This debate only abated in the past years once alternative psychological models took hold, which were based in part on insights from results associated with functional imaging (Wais et al., 2010, J of Cognitive Neuroscience; Wais, 2013, J Neurophysiology).
Beyond the unnecessarily narrow definitions of recollection and familiarity in the Introduction and Discussion (i.e., lines 91-93, 96-97, 415-416, 429-437, etc.), interpretation of the results between PDP and TDP is undermined by the flaw that memory strength associated with putative recollection responses is not equated with memory strength associated with putative familiarity responses (Slotnick & Dodson, 2005, Memory & Cognition; Wixted, 2007, Psychological Review; Wais et al., 2008, J Cognitive Neuroscience). As such, there is a confound of memory strength and memory processes/qualities in the reported data. This confounding analysis substantially diminishes the impact of the manuscript’s main finding as it is presented in terms of the HTDP model.
An alternative approach would be for the authors to characterize their experiments as a comparison between generalized recognition memory (PDP) and hippocampal-dependent associative memory (TDP) with NS youth. Indeed, this view would be consistent with the foundation laid out in Introduction lines 1-84. As foreshadowed above with just a few of the many available references, the notion of a clear division of labor in the MTL, as the authors relate in lines 95-109, has been debunked in a series of fMRI studies. Taking into account that hippocampal function is indeed associated with memory strength, as well as the several qualities of high-fidelity LTM (Wais, 2011, Hippocampus), the submitted manuscript could improve the clarity of its findings through a more balanced view of how MTL and its multi-stage development support the particular LTM processes diminished in NS (see Canada et al., 2019, Cerebral Cortex).
- Specifically, results for PDP between groups (Table 2) show a pattern for the exclusion condition that is to be expected where an analysis of confidence indicated with each response is lacking (i.e., for both hits and false alarms). Also, analysis for false alarm rates here would be more informative than for hit rates because correct rejections of lures provide a more precise estimate for hippocampal-dependent LTM than recognition of targets. On its face, the finding of no differences between NS and TD on the PDP suggests that both groups are capable of the same level of recognition memory strength, which is an appealing point of support for the authors’ eventual conclusion that NS youth suffer a particular deficit in hippocampal-dependent associative memory.
Moreover, the lengthy explications of the HTDP analyses in lines 351-376 could well be eliminated if a simpler study-source memory approach were applied here with these data (Slotnick and Dodson, 2005, Memory & Cognition), which operates in essentially the same manner as PDP.
Finally, the results for TDP between groups (Table 3) can be very well explained by the interpretation that hippocampal-dependent associative memory is selectively impaired in NS youth, relative to TD.
- LTM development at 8-9 years, behaviorally and in terms of brain functionality, is immature. The Introduction spends three lengthy paragraphs reviewing the effects of NS on cellular signaling in the hippocampus, but no relationship is proposed for this background relative to healthy development of LTM through childhood and young adulthood. As noted above, neither the relevant literature, nor the PDP results here in Table 2, support the notion that the exclusive function of the hippocampus is to support associative or other qualities of high-fidelity LTM.
The Introduction should address how healthy LTM function develops before and after prefrontal white matter maturation, as well as what healthy hippocampal contributions are at these stages. It appears an over-simplification to suggest that because hippocampal signaling in 8-year-old NS is at a deficit, then average high-fidelity LTM capability will not catch up by the age that hippocampal-prefrontal connectivity is fully developed (or even will it develop?).
Minor points:
- Analysis and calculation of the repeated measures ANCOVA assumed equal variance models (line 317). Distributions of recognition confidence for targets and lures in LTM are not equally distributed (Wixted, 2007). Dispensing with the gymnastics of the parameter estimates required by the HTDP (lines 351-370) could address this problem.
- There appears little connection to the present behavioral study for the lengthy Discussion of normal adult functional and structural networks subserving LTM (lines 476-531). The detail in this material does not seem relevant to findings in the report. Consideration of how intra-hippocampal development disorders might affect eventual maturation and functionality of the normal hippocampal-cortical memory system would be of interest for both the Introduction and Discussion.
Author Response
Response to Reviewer 1 Comments
Point 1: The manuscript reports a study that examined long-term memory (LTM) deficits in Noonan Syndrome (NS) youth, relative to normally developing youth. The mean ages of the NS and healthy groups were approximately 8 to 9 years of age, meaning that the experiments assessed LTM capability of the participants well before maturation of their functional hippocampal-cortical brain networks, which are necessary for the full fidelity in LTM retrieval (see Ofen et al., 2007, Nature Neuroscience).
The experiment groups are appropriately characterized, the methods well explained and the results clear. The Introduction of the study and Discussion of the findings are useful, if quite lengthy. The English translation can improve in syntax and noun-verb agreement. The two LTM measures are appropriate for the authors’ motivation to assess the participants’ high-fidelity LTM capability, but the authors’ interpretations according to the Yonelinas high-threshold dual process (HTDP) model do not serve their findings adequately.
Rather than struggle with the known and demonstrated limitations of the HTDP model beheld in the text, the report could bring more valuable insight to the literature about NS LTM deficits if it were to draw a more balanced view of cognitive models in high-fidelity LTM. Three major and a few minor concerns are explained below.
Response 1: We thank this Reviewer for his/her valuable suggestion. We agree with this Reviewer that the Yonelinas high-threshold dual process (HTDP) model do not serve all our findings adequately and we have moved to a more balanced view of cognitive models for the interpretation of our data, taking into account the memory strength hypotheses (previously known as Single Trace Theory) (Evans and Wilding, 2012; Wais et al., 2013). In particular, in the interpretation of our data as expressions of recollection and familiarity processes, both the prevalent role of the hippocampus in recollection (as suggested by the Dual process theories) and a more widespread involvement of MTL cortices in both recollection and familiarity (as proposed by the strength hypotheses) have been considered (see in particular substantially modified Introduction and completely rewritten second half of the Discussion). Moreover, alternative to the theoretical framework of the recollection and familiarity, an interpretation of our data as expressions of deficits in source memory and associative recognition in children with NS has been reported (see the paragraph in the Discussion page 14).
Accordingly, we also decided to modify the title in “Recognition memory in Noonan Syndrome” avoiding to underline the presence of a clear dissociation between the two components of familiarity and recollection.
As you can see, we have substantially modified the entire manuscript moving towards a more balanced modeling for the interpretation of our memory data. English syntax and noun-verb agreement was also improved.
Point 2:
The manuscript refers to recollection and familiarity, from line 2 through line 546, as though the model for these catch-all memory process labels were well defined and accepted in the relevant literature. Quite to the contrary, a hot controversy about the substantial limitations of HTDP has roiled the field (Slotnick & Dodson, 2005, Memory & Cognition; Wixted, 2007, Psychological Review; Wais, 2008, Neuropsychologia). This debate only abated in the past years once alternative psychological models took hold, which were based in part on insights from results associated with functional imaging (Wais et al., 2010, J of Cognitive Neuroscience; Wais, 2013, J Neurophysiology).
Response 2: Following the Reviewer’s suggestion we have now largely explained the alternative psychological models in introduction and in the discussion and took in account a new balance view for the interpretation of results. Specifically changes can be found on:
Page 2-3
In the last two decades, the dichotomy between recollection and familiarity experiences has been one of the most investigated issues in human declarative episodic memory. In recollection, exposure to a previously encountered item cues the retrieval of a number of contextual, temporal and other associative information that makes the experience of retrieval vivid, such as reliving the event in the mind’s eye. In familiarity, however, the item is recognized as having been previously encountered but does not cue any associative information, so the act of remembering is just a feeling of a previous encounter. The different characteristics of the two memory experiences make performance on recall tasks entirely dependent on recollection; conversely, recognition memory would be subsumed by either recollection or familiarity, the relative contribution of the two memory components depending at least partially on the associative vs the single-item nature of the memorandum [23-25].
At a theoretical level, a critical question is what process or processes underlie these two experiences. According to dual-process theories [26,27], episodic retrieval depends on two independent processes: familiarity and recollection.
In contrast, according to strength theories, episodic retrieval reflects a single memory strength dimension. A recent strength theory, the aggregate-strength model, incorporates the familiarity–recollection distinction by assuming that both processes are graded and are aggregated into a single strength signal before the retrieval decision [28-34]. More recently, The Complementary Learning Systems (CLS) model proposes that recognition based on pattern completion is experienced as a generalized match with some prior experience, whereas recognition accompanied by awareness of relevant contextual details is based on pattern separation [35-36].
Page 14.
….The alternative theoretical position relative to the nature of memory processes involved in familiarity and recollection assumes, instead, that episodic retrieval reflects a single memory strength dimension. As for the neural circuits implicated, this theory does not posit a differential involvement of mesial temporal lobe structures in recollection and familiarity, since both the hippocampus and perirhinal cortex would be implicated in the memory strength detection underlying the two components [31-33, 68].
Following the memory strength hypotheses of recollection and familiarity, a widespread involvement of mesiotemporal structures, involving both the hippocampus and parahippocampal cortices, in individuals with NS is expected. Johnson et al. [66] did not report the exact localization of atrophic changes in the parahippocampal gyrus of NS individuals. Fattah et al. [69] found that children with NS exhibited significant decreases in volume and thickness of striatum and regions surrounding the hippocampus.Specifically,the bilateral entorhinal cortex exhibited significant decreases as did the right temporal pole surface area and the right parahippocampal gyrus.”
Point 3: Beyond the unnecessarily narrow definitions of recollection and familiarity in the Introduction and Discussion (i.e., lines 91-93, 96-97, 415-416, 429-437, etc.), interpretation of the results between PDP and TDP is undermined by the flaw that memory strength associated with putative recollection responses is not equated with memory strength associated with putative familiarity responses (Slotnick & Dodson, 2005, Memory & Cognition; Wixted, 2007, Psychological Review; Wais et al., 2008, J Cognitive Neuroscience). As such, there is a confound of memory strength and memory processes/qualities in the reported data. This confounding analysis substantially diminishes the impact of the manuscript’s main finding as it is presented in terms of the HTDP model.
An alternative approach would be for the authors to characterize their experiments as a comparison between generalized recognition memory (PDP) and hippocampal-dependent associative memory (TDP) with NS youth. Indeed, this view would be consistent with the foundation laid out in Introduction lines 1-84. As foreshadowed above with just a few of the many available references, the notion of a clear division of labor in the MTL, as the authors relate in lines 95-109, has been debunked in a series of fMRI studies. Taking into account that hippocampal function is indeed associated with memory strength, as well as the several qualities of high-fidelity LTM (Wais, 2011, Hippocampus), the submitted manuscript could improve the clarity of its findings through a more balanced view of how MTL and its multi-stage development support the particular LTM processes diminished in NS (see Canada et al., 2019, Cerebral Cortex).
Response 3: As previously noted, partially following the reviewer’s suggestion, we have now added in the Discussion section a completely new paragraph in which an interpretation alternative to the recollection/familiarity theoretical framework of the performance of NS subjects in the PDP and TDP tasks is provided. We have indeed acknowledged that both the PDP and the TDP tasks involve a condition of single item recognition and another condition emphasizing associative recognition and, in this sense, the results of this study could also be interpreted as reflecting a dissociation in NS individuals between preserved single item but reduced associative memory. In the light of the rich literature attributing a prevalent role to hippocampal neural circuits in binding information in long-term memory, the results of this could still be suggestive of a prevalent involvement of hippocampal functions, even outside the assumptions of the dual-process theory.
See changes in introduction:
“The aim of the current study was to provide a detailed characterization of recognition memory of children and adolescents with NS and the clinically related NSML. To this end, we used two different experimental paradigms which more directly assess the contribution of familiarity and recollection on recognition memory tasks. The first was a modified version of the Process Dissociation Procedure (PDP) devised by Jacoby [53] and adjusted for a developmental population by Costanzo et al. [52]. In this paradigm the recollection and familiarity components of recognition are estimated by contrasting different recognition conditions for the studied material [53-55] under different source memory conditions. The second was a Task Dissociation Procedure (TDP), where performance on tests of single item recognition, which reflect familiarity strengths, and associative item recognition, which requires retrieval of contextual information involved in recollection, are compared.
Consistent with previous evidence of lower performance on recall than recognition tasks, we expected that individuals with NS recollection would be more affected on recollection than familiarity components of recognition memory. “
“
And in discussion
An interpretation of the data reported in the present study, alternative to the conceptual framework of recollection and familiarity, is one which distinguishes identifies in both procedures proposed an experimental condition assessing memory skills for single information and another evaluating associative memory. The PDP could be considered a test which, in the condition of exclusion, investigates source memory skills, a type of associative memory in which the correct recollection of the context is the necessary prerequisite for adequate performance. The task dissociation procedure, in turn, explicitly contrasts a condition of recognition of single items and a condition of recognition of between-items associations. There is general consensus in the literature that associative memory recruits hippocampal resources whereas single item recognition can be accomplished by parahippocampal cortices [70-71]. There is, in particular, compelling evidence of a peculiar role of hippocampus in source memory (for a review, see 72). In this sense, therefore, the data of the present study could still be suggestive of a prevalent involvement of hippocampal functions, even outside the assumptions of the dual-process theory.
Point 4: Specifically, results for PDP between groups (Table 2) show a pattern for the exclusion condition that is to be expected where an analysis of confidence indicated with each response is lacking (i.e., for both hits and false alarms). Also, analysis for false alarm rates here would be more informative than for hit rates because correct rejections of lures provide a more precise estimate for hippocampal-dependent LTM than recognition of targets. On its face, the finding of no differences between NS and TD on the PDP suggests that both groups are capable of the same level of recognition memory strength, which is an appealing point of support for the authors’ eventual conclusion that NS youth suffer a particular deficit in hippocampal-dependent associative memory.
Response 4: We thank the Reviewer for the observation. Analysis on false alarm rates on both the inclusion and exclusions conditions for new items did not result in any significant effect, in line with the assumptions underlying the PDP.
See page 11
“To establish whether assumptions underlying the PDP were met, it was first examined whether acceptance rates to new items differed between inclusion and exclusion across groups. The proportion of “new” responses for the colored drawings and spoken words were analyzed by means of repeated measures ANCOVA with Group (NS vs TD) as a between factor and Condition (inclusion vs exclusion) as within factors, and LQ and CA as covariates. Neither the Group (F1,36=<.001; p=.99, pη2<.01), the Condition (F1,36=1.62; p=.21, pη2=.04) and the interaction between Group and Condition (F1,30=.76; p=.39, pη2=.02) resulted significant.”
Conversely, NS resulted significantly poorer than TD children in the rate of correct endorsements in the inclusion and in the difference between correct endorsement in the inclusion and incorrect endorsement of non target in the exclusion condition. Partially following the reviewer’s suggestion we recalculated the estimate of recollection and familiarity with the calculation formula of Jacoby, 1999, further applied by Koenig et al., 2015, 2020 to test single single-process account of developmental trends in recognition using PDP.
See results:
Recollection was calculated by subtracting “old” responses to non-targets (spoken words) in the exclusion condition from “old” responses to non-targets in the inclusion condition, R = “old” responses inclusionnontarget – “old” responses exclusionnontarget. Familiarity was calculated from estimates of recollection, F = “old” responses exclusion non-target /(1 – R) (see 53-57).
The one-way ANCOVAs revealed no significant difference among groups for the familiarity estimate F (participants with NS/NMLS=.24 ± .30; TD=.26 ± .36; F1,36=.52, p=.48, pη2=.01), but TD displayed a significantly higher recollection estimate R (.88 ± .13) than participants with NS/NMLS (.72 ± .22; F1,36=5.24, p=.03, pη2=.13).
Moreover, we did not discuss our data as reflecting a dissociation in NS between procedures (spared performance on the PDP but impaired on the TDP) but rather as reflecting a dissociation between conditions inside each task (interpreted as the expression of spared familiarity but poor recollection or, alternatively, as spared single item but poor associative recognition). See page 14 of the discussion.
Point 5: Moreover, the lengthy explications of the HTDP analyses in lines 351-376 could well be eliminated if a simpler study-source memory approach were applied here with these data (Slotnick and Dodson, 2005, Memory & Cognition), which operates in essentially the same manner as PDP.
Response 5: Again, we thank the Reviewer for the observation. According to his/her suggestion, we have now removed the lengthy explications of the HTDP analyses estimate of recollection and familiarity from PDP results and recalculation the estimate with the formula of Jacoby, 1999, further applied by Koenig et al., 2015, 2020 to test single single-process account of developmental trends in recognition using PDP.
Point 6: Finally, the results for TDP between groups (Table 3) can be very well explained by the interpretation that hippocampal-dependent associative memory is selectively impaired in NS youth, relative to TD.
Response 6: As previously noted, we agree with the Reviewer’s suggestion that an interpretation of our data alternative to that deriving from the recollection/familiarity theoretical framework and still consistent with and hippocampal-dependent deficit of recognition memory is possible. We have discuss this important issue in a completely new paragraph of the Discussion section.
Please find changes all over the Discussion, from page 14 to 16:
Point 7: LTM development at 8-9 years, behaviorally and in terms of brain functionality, is immature. The Introduction spends three lengthy paragraphs reviewing the effects of NS on cellular signaling in the hippocampus, but no relationship is proposed for this background relative to healthy development of LTM through childhood and young adulthood. As noted above, neither the relevant literature, nor the PDP results here in Table 2, support the notion that the exclusive function of the hippocampus is to support associative or other qualities of high-fidelity LTM.
Response 7: Following the reviewer’s suggestion, we have now reduced in the Introduction the paragraph on cellular signaling and animal modeling and have added the necessary background relative to healthy development of LTM through childhood and of other regions than hippocampus supporting associative or other qualities of high-fidelity LTM in both the Introduction and the Discussion section.
See introduction:
From a developmental perspective, there is documented evidence of age-related increases on tasks of memory recognition that require few contextual details and protracted development (i.e., gains continuing after middle childhood) then on tasks that require retention of detailed information [31-44]. Researchers have highlighted the importance of related cognitive functions, as working memory, metacognition and mnemonic strategies [45], in explaining individual differences in memory performance. Neuroimaging studies have shown neural basis of age-related gains in memory performance. Differential developmental trajectories in the prefrontal cortex (PFC) and medial temporal lobe (MTL), including the hippocampus [42-43], have been linked to memory outcomes in associative memory test performance [46-47] and source memory performance [48-48]. Moreover, age-related differences in functional connectivity between PFC and MTL seam to account for age-related increases in memory performance [50].
See discussion:
“Moreover, given the well recognized age-related differences in multiple hippocampal subfield volumes in the age range of our patients (Lee et al. 2014; Tamnes et al. 2014; Daugherty et al. 2017; Riggins et al. 2018), we considered our results taking into account possible change related to ages.
“Many investigators have proposed that the lack of detailed memory for early childhood memories is due to the protracted development of the hippocampus [63-64]. Moreover, recent studies have shown age-related differences in multiple hippocampal subfield volumes from 4 to 13 years of age [46-47, 65]. “
“A slower development of memory for context has been associated with the maturation of PFC functioning [43]. Postmortem and structural imaging evidence indicates that PFC maturation appears to continue into late adolescence [50,73].”
Point 8: The Introduction should address how healthy LTM function develops before and after prefrontal white matter maturation, as well as what healthy hippocampal contributions are at these stages. It appears an over-simplification to suggest that because hippocampal signaling in 8-year-old NS is at a deficit, then average high-fidelity LTM capability will not catch up by the age that hippocampal-prefrontal connectivity is fully developed (or even will it develop?).
Response 7: Following the Reviewer’s suggestion we have now addressed in the introduction and commented in the discussion the role of the maturation of LTM and PFC structures in recognition abilities.
See Discussion
A reduction in cortical thickness was also reported in middle frontal areas in children with NS [66,69]. One could hypothesize that abnormalities in the connectivity of the cortical regions adjacent to the hippocampus (with frontal and hippocampal cortices), rather than their integrity, might explain the memory profile of impaired associative recognition and preserved general memory strenghts in our group of children and adolescents with NS/NSML.
A slower development of memory for context has been associated with the maturation of PFC functioning [43]. Postmortem and structural imaging evidence indicates that PFC maturation appears to continue into late adolescence [50,73]. For this reason, the role of reduced prefrontal white matter maturation in children with NS, as well as reduced hippocampal contributions, may account for the associative memory deficits found in our group of children with NS/NSML. Taken together, findings of age differences in PFC activation patterns suggest that the development of attentional and strategic control processes involved in the encoding and retrieval of detailed representations of experiences is key to predicting age-related differences in memory performance.
Minor points:
Point 9: REV: 4 Analysis and calculation of the repeated measures ANCOVA assumed equal variance models (line 317). Distributions of recognition confidence for targets and lures in LTM are not equally distributed (Wixted, 2007). Dispensing with the gymnastics of the parameter estimates required by the HTDP (lines 351-370) could address this problem.
Response 9: We thank the Reviewer, however, recognition confidence for targets and lures were not collected in our PDP. Indeed, although the modelling of confidence ratings is common in research with adults, the assumption that children at different age groups indicate their confidence as accurately as adults is an open question and increases with age (Roebers et al., 2004).
As these models cannot be applied to our current data because we have not collected confidence ratings, we considered the difference between old-new responses to calculate familiarity and recollection, considering the correct endorsement in the inclusion and incorrect endorsement of non target in the exclusion condition. The formula of Jacoby, 1999 was used as applied by Koenig et al., 2015, 2020 to test single single-process account of developmental trends in recognition using PDP.
Point 10: There appears little connection to the present behavioral study for the lengthy Discussion of normal adult functional and structural networks subserving LTM (lines 476-531). The detail in this material does not seem relevant to findings in the report. Consideration of how intra-hippocampal development disorders might affect eventual maturation and functionality of the normal hippocampal-cortical memory system would be of interest for both the Introduction and Discussion.
Response 10: Following the reviewer’s suggestion, we have removed the discussion of anatomical and functional networks subserving adult normal LTM. As previously noted, we have inserted instead a discussion of how the normal maturation of these structures may inform about possible developmental disorders. See text paragraphs in response 6 and 7.
Reviewer 2 Report
The authors explored recognition memory in children with Noonan syndrome and a group of age-matched controls. To estimate the contribution of familiarity and recollection a process-dissociation procedure (PDP) was used. Also, a separate task contrasted item memory (mainly relying on familiarity) and associative memory (mainly relying on recollection). The findings suggest that NS group shows intact familiarity but impaired recollection relative to controls. Implications for brain circuitry and rehabilitation protocols are discussed.
I have previously reviewed the paper for another journal. I was and remain positive about the overall scope and importance of the findings, but there are still a few issues that require addressing and a few limitations that need to be taken into account. Many of this points were raised in my previous review, but the authors never had the chance to respond to these (and they haven’t been addressed in the revised manuscript).
1) Age of the participants: patients’ age range varies a lot, so inevitably grouping these patients together undermines the validity of the findings. One suggestion will be to add age as a covariate in all the analyses. Also, please add the ages of each participant in Table 1, so we can appreciate how many belonged to younger and how many to older age groups.
2) The task dissociation task is based, in my opinion, on very few retrieval trials i.e., 3 per condition. Perhaps the authors need to justify this selection and acknowledge the limitation. Also, this is relevant in the interpretation of the ceiling effect in the item memory condition. The findings there do not necessarily mean there are no differences between the two groups especially when the data are based on only 3 trials.
3) Related to this observation, the comparison of familiarity in the PDP task, although not significantly different between patients and controls is towards the same direction as the difference observed for recollection. Therefore, this does not justify the discussion of a clear dissociation between F&R in this sample.
4) It’s not very clear from the description whether similarity of foils and targets in the TPT task was taken into account. In the figure, the second trial of the item recognition task includes very similar foils (sunflowers), whereas trial 3 includes a more distinctive target (relative to the foils). Was this something that was manipulated in the experiment? The degree of similarity may have great impact and affect the actual reliance on familiarity versus recollection in item recognition.
5) Did the authors test homogeneity of variance in the different analyses. This is important considering the low number of patients in the sample.
6) The discussion of the role of the parahippocampal cortex in F/R remains very limited in the paper and is based on two review papers from 2007. More recent studies (published over the last few years) have explored the role of the parahippocampal gyrus in F/R also in relation to stimulus type and therefore would be more appropriate for the discussion.
7) “Row data from the PDP…” is Row a typo here?
Author Response
Response to Reviewer 2 Comments
Point 1: The authors explored recognition memory in children with Noonan syndrome and a group of age-matched controls. To estimate the contribution of familiarity and recollection a process-dissociation procedure (PDP) was used. Also, a separate task contrasted item memory (mainly relying on familiarity) and associative memory (mainly relying on recollection). The findings suggest that NS group shows intact familiarity but impaired recollection relative to controls. Implications for brain circuitry and rehabilitation protocols are discussed.
I have previously reviewed the paper for another journal. I was and remain positive about the overall scope and importance of the findings, but there are still a few issues that require addressing and a few limitations that need to be taken into account. Many of this points were raised in my previous review, but the authors never had the chance to respond to these (and they haven’t been addressed in the revised manuscript).
Response 1: We are very grateful to the Reviewer for his/her time and valuable suggestion. We hope that, having the opportunity to review our paper, we have now substantially and significantly improved our manuscript.
Point 2: Age of the participants: patients’ age range varies a lot, so inevitably grouping these patients together undermines the validity of the findings. One suggestion will be to add age as a covariate in all the analyses. Also, please add the ages of each participant in Table 1, so we can appreciate how many belonged to younger and how many to older age groups.
Response 1: We thank the Reviewer for the valuable suggestion to include age as a covariate in all the analyses. Given the well recognized age-related differences in multiple hippocampal subfield volumes in the age range of our patients, taking into account possible change related to ages in our results is of great relevance.
Since CA had a skewed distribution, data were normalized by by Box–Cox transformation (see graph 1 below).
We have now added age as a second covariate in all the analysis. Results do no changes substantially, but significantly confirmed the results. See all the results.
Box–Cox transformation of Chronological age (CA)
|
Subject |
Sex |
Age |
IQ |
LQ |
Disorder |
Gene |
Amino acid change |
|
1 |
M |
6.08 |
90 |
84 |
NS |
SOS1 |
Gly434Arg |
|
2 |
F |
13 |
90 |
90 |
NS |
PTPN11 |
Asn308Ser |
|
3 |
F |
6.75 |
120 |
98 |
NS |
PTPN11 |
Thr42Ala |
|
4 |
F |
8.58 |
100 |
106 |
NS |
SOS1 |
Ile733Phe |
|
5 |
M |
6.42 |
120 |
84 |
NS |
PTPN11 |
Ala72Ser |
|
6 |
M |
10.08 |
102 |
94 |
NS |
RIT1 |
Ser35Thr |
|
7 |
F |
12.5 |
80 |
95 |
NS |
SOS1 |
Arg552Lys |
|
8 |
M |
13.75 |
100 |
111 |
NS |
PTPN11 |
Met504Thr |
|
9 |
F |
5.75 |
90 |
84 |
NS |
SOS1 |
Arg552Ser |
|
10 |
M |
6.25 |
110 |
89 |
NS |
PTPN11 |
E139D |
|
11 |
F |
5.75 |
130 |
102 |
NS |
SOS1 |
Glu433Lys |
|
12 |
M |
7.33 |
100 |
71 |
NS |
PTPN11 |
Asn308Ser |
|
13 |
F |
5.41 |
110 |
74 |
NS |
RAF1 |
S257L |
|
14 |
F |
7.66 |
90 |
73 |
NS |
PTPN11 |
Phe285Ser |
|
15 |
M |
13.08 |
120 |
80 |
NS |
PTPN11 |
Gly503Arg |
|
16 |
F |
7.42 |
101 |
120 |
NSML |
PTPN11 |
Tyr279Ser |
|
17 |
F |
12.33 |
125 |
122 |
NSML |
PTPN11 |
Thr468Met |
|
18 |
M |
9.08 |
120 |
108 |
NSML |
PTPN11 |
Thr468Met |
Moreover, as suggested, we have now provided the age of participants in Table 1. (see table 1)
TAB 1
Point 3: The task dissociation task is based, in my opinion, on very few retrieval trials i.e., 3 per condition. Perhaps the authors need to justify this selection and acknowledge the limitation. Also, this is relevant in the interpretation of the ceiling effect in the item memory condition. The findings there do not necessarily mean there are no differences between the two groups especially when the data are based on only 3 trials.
Response 2: We agree with the Reviewer and have acknowledged the limitation in the Discussion.
See page 14:
“We acknowledge that the virtually perfect performance of the two groups in the single item recognition task of the TDP (mean percentage of correct responses ≥ 0.95 for each of them) may represent a limitation in revealing a clear cut dissociation between groups in this task. Specifically, the presence of only three retrieval trials may be prevent the possibility to detect possible difference between participants in the single item task.”
Point 4: Related to this observation, the comparison of familiarity in the PDP task, although not significantly different between patients and controls is towards the same direction as the difference observed for recollection. Therefore, this does not justify the discussion of a clear dissociation between F&R in this sample.
Response 4: We agree with the present reviewer and also following the first reviewer’s suggestion we acknowledge that the clear dissociation do not serve all our findings adequately. We have then moved to a more balanced view of cognitive models for the interpretation of our data, taking into account both the Single Trace Theory and the more integrated Complementary Learning Systems (CLS) model (Evans and Wilding, 2012; Wais et al., 2013). In this more balanced view, hippocampal deficit could be considered as a possible explanation for the associative memory deficits within a relative preserved general memory strengths. We have almost re-written the introduction and the discussion and modified abstract accordingly. See also response 1, 2 and 3 to Reviewer 1.
Accordingly, we also decided to modify the title in “Recognition memory in Noonan Syndrome” avoiding to underline the presence of a clear dissociation between the two components of familiarity and recollection.
See also in the Discussion:
“An interpretation of the data reported in the present study, alternative to the conceptual framework of recollection and familiarity, is one which distinguishes identifies in both procedures proposed an experimental condition assessing memory skills for single information and another evaluating associative memory. The PDP could be considered a test which, in the condition of exclusion, investigates source memory skills, a type of associative memory in which the correct recollection of the context is the necessary prerequisite for adequate performance. The task dissociation procedure, in turn, explicitly contrasts a condition of recognition of single items and a condition of recognition of between-items associations. There is general consensus in the literature that associative memory recruits hippocampal resources whereas single item recognition can be accomplished by parahippocampal cortices [70-71]. There is, in particular, compelling evidence of a peculiar role of hippocampus in source memory (for a review, see 72). In this sense, therefore, the data of the present study could still be suggestive of a prevalent involvement of hippocampal functions, even outside the assumptions of the dual-process theory.”
“By using two independent and direct experimental paradigms, our study shows a reduced contribution of recollection in the recognition memory performance of a group of children and adolescents with NS and the clinically related NSML. The impairment of detailed memory or recollection in children with RASophaties is direct evidence supporting the neurobiological hypothesis that RAS–MAPK mutations can disrupt hippocampal and temporal lobe memory formation and consolidation. The present findings provide insights into the declarative memory profile of these individuals,…”
Point 5: It’s not very clear from the description whether similarity of foils and targets in the TPT task was taken into account. In the figure, the second trial of the item recognition task includes very similar foils (sunflowers), whereas trial 3 includes a more distinctive target (relative to the foils). Was this something that was manipulated in the experiment? The degree of similarity may have great impact and affect the actual reliance on familiarity versus recollection in item recognition.
Response 5: As correctly underlined by the Revewer’s, the degree of similarity may have great impact and affect the actual reliance on familiarity versus recollection in item recognition. We controlled and counterbalanced between trials the distinctiveness of target relative to the foils for specific flowers, considering also the background. All stimuli were selected for distinctiveness between a pool of items in a pilot study. Details of the method have been published in Costanzo et al., 2013.
Point 6: Did the authors test homogeneity of variance in the different analyses. This is important considering the low number of patients in the sample.
Response 6: We have tested the homogeneity of variance in the different analyses. In particular, Levene’s test was used to check for homogeneity and Mauchly’s test was used to check for a violation of sphericity. The overall level of significance was set at p < 0.05. The Welch test was performed on demographic characteristics comparisons when the homogeneity of variance of the data was invalid. If a violation of sphericity was detected, a Greenhouse–Geisser correction was applied. In the ANCOVAs analyses, since CA had a skewed distribution, data were normalized by by Box–Cox transformation.
We have now better specified the method.
Point 7: The discussion of the role of the parahippocampal cortex in F/R remains very limited in the paper and is based on two review papers from 2007. More recent studies (published over the last few years) have explored the role of the parahippocampal gyrus in F/R also in relation to stimulus type and therefore would be more appropriate for the discussion.
Response 7: We have substantially rewrite the discussion on the brain regions hypothesis and have specified the importance of parahippocampal gyrus for perceptual identity of the stimulus and in object-object or object-context relational processing, which is relevant to our TDP task.
See Discussion, page 17-18
“Parahippocampal cortices, including parahippocampal place area and the fusiform gyrus, have been also involved in visual recognition processing, specifically in object-object or object-context relational processing [76-77]. Given the material used in the TDP was only pictorial and required object-object relational recognition, it cannot be excluded that the associative recognition deficit, exhibited by children with NS, could be partly due to perceptual deficits per se, related to parahippocampal abnormalities. Previous findings in NS and other RASopathies have indeed displayed better performance on verbal than on visual memory tasks [20,22]. The fact that children with NS have also shown source memory deficits in the PDP, that included words or easily verbalizable pictures, suggests a more generalized deficit in detailed episodic memory in NS population, beyond the modality of the memorandum.”
“Indeed, manipulating the material to be learned in order to increase the reliance on familiarity (spared) rather than associative memory (impaired) (e.g., by increasing consistency of the perceptual format of stimuli during study and testing) or by repeating or spacing learning sessions, as demonstrated in animal models [84], could improve learning therapy for individuals with NS.”
Point 8: “Row data from the PDP…” is Row a typo here?
Response 8: Corrected.

Round 2
Reviewer 1 Report
The breadth of the revision and supplemental references are satisfactory.
Author Response
We would like to thank the reviewer for his/her valuable and constructive suggestions and for the positive comments on our revised manuscript.
Best regards,
Floriana Costanzo and co-authors.
Reviewer 2 Report
I am generally satisfied with the changes made to the manuscript in relation to the issues raised in the previous round of review. However, I outline below three points in relation to the Discussion, which would require some minor revisions.
- As the authors discuss extensively the role of the hippocampus, parahippocampal gyrus and the interaction with stimulus type, I think it’s very relevant to consider empirical evidence from Kafkas et al 2017 Hippocampus, 27(2), 194–209. https://doi.org/10.1002/hipo.22683
- Line 485-486: “…is one which distinguishes identifies in both procedures proposed an experimental condition…”, the sentence is unclear.
- Line 493-496: “There is, in particular, compelling evidence of a peculiar role of hippocampus in source memory (for a review, see 72). In this sense, therefore, the data of the present study could still be suggestive of a prevalent involvement of hippocampal functions, even outside the assumptions of the dual-process theory.”. This section is very vague and I do not think it adds anything relevant to the discussion. First of all, the adjective “peculiar” to characterise the role of the hippocampus is strange and perhaps it’s a typo. Also, the next sentence is rather cryptic; what this role of the hippocampus might be that is not compatible with its supposed role in recollection/source memory? If the authors imply that dual-process models support exclusive role of the hippocampus in recollection and not in other functions such as source memory, the premise is definitely wrong. As the authors are aware, the role of the hippocampus in recollection as seen by dual-process theorists is compatible to its role in processes/mechanisms such as recall, source memory retrieval, associative processing, pattern completion etc. Therefore, what the authors refer to here is not clear.
Author Response
We thank the reviewer for his/her careful reading of the manuscript and constructive remarks. We have taken the comments on board to improve and clarify the revised manuscript. Please find below a detailed point-by-point response to the last comments (reviewers’ comments in black, our replies in bold)
Point 1: As the authors discuss extensively the role of the hippocampus, parahippocampal gyrus and the interaction with stimulus type, I think it’s very relevant to consider empirical evidence from Kafkas et al 2017 Hippocampus, 27(2), 194–209. https://doi.org/10.1002/hipo.22683
RESP 1: We thank you the reviewer for this valuable suggestion. We have now included the reference to this valuable paper to comment the role of MTL structures on recognition and the interaction with stimulus type.
See Discussion lines 572-577:
The fact that children with NS have also shown source memory deficits in the PDP, that included words or easily verbalizable pictures, suggests a more generalized deficit in detailed episodic memory in NS population, beyond the modality of the memorandum. This is in line with fMRI evidence [Kafkas 2017] who found degree of specialization, with respect to stimulus type, within the MTL structures. Indeed, different parahippocampal cortices underlayunderlie familiarity, for different kinds of visual stimuli, particularly while parahippocampus for visual stimuli, while regions of the hippocampus mediateded recollection, regardless of the kind of stimulus.
Point 2: Line 485-486: “…is one which distinguishes identifies in both procedures proposed an experimental condition…”, the sentence is unclear.
RESP 2: We have now rephrased the sentence to be more clear.
“The hypothesis of greater involvement of the hippocampus in the recognition memory deficits of individuals with NS is, however, strengthened by the specific low performance showed by our participants with NS in the experimental conditions assessing associative recognition. Indeed, the PDP could be considered a test which, in the condition of exclusion, investigates source memory skills, a type of associative memory in which the correct recollection of the context is the necessary prerequisite for adequate performance. The task dissociation procedure, in turn, explicitly contrasts a condition of recognition of single items and a condition of recognition of between-items associations.”
Point 3: Line 493-496: “There is, in particular, compelling evidence of a peculiar role of hippocampus in source memory (for a review, see 72). In this sense, therefore, the data of the present study could still be suggestive of a prevalent involvement of hippocampal functions, even outside the assumptions of the dual-process theory.”. This section is very vague and I do not think it adds anything relevant to the discussion. First of all, the adjective “peculiar” to characterise the role of the hippocampus is strange and perhaps it’s a typo. Also, the next sentence is rather cryptic; what this role of the hippocampus might be that is not compatible with its supposed role in recollection/source memory? If the authors imply that dual-process models support exclusive role of the hippocampus in recollection and not in other functions such as source memory, the premise is definitely wrong. As the authors are aware, the role of the hippocampus in recollection as seen by dual-process theorists is compatible to its role in processes/mechanisms such as recall, source memory retrieval, associative processing, pattern completion etc. Therefore, what the authors refer to here is not clear.
RESP 2: We acknowledge that the concept has not very clearly expressed in this pargraph, although we completely agree with the referee's view. We have now reframed all the paragraph hoping to have better clarified the interpretation.
See all the Discussion paragraph lines 493-507 and accordingly the final sentence:
“In particular, compelling evidence has been provided of a relevant role of hippocampus in source memory as well as recall and associative memory [for a review, see 72].”